# Blind Source Separation Method Based on Neural Network with Bias Term and Maximum Likelihood Estimation Criterion

**DOI:** 10.3390/s21030973

**Published:** 2021-02-01

**Authors:** Sheng Liu, Bangmin Wang, Lanyong Zhang

**Affiliations:** College of Intelligent Systems Science and Engineering, Harbin Engineering University, Harbin 150001, China; liu.sch@163.com (S.L.); zlyalf@sina.com (L.Z.)

**Keywords:** blind source separation, feedforward neural network, maximum likelihood estimation, gradient optimization algorithm

## Abstract

Convergence speed and steady-state source separation performance are crucial for enable engineering applications of blind source separation methods. The modification of the loss function of the blind source separation algorithm and optimization of the algorithm to improve its performance from the perspective of neural networks (NNs) is a novel concept. In this paper, a blind source separation method, combining the maximum likelihood estimation criterion and an NN with a bias term, is proposed. The method adds L2 regularization terms for weights and biases to the loss function to improve the steady-state performance and designs a novel optimization algorithm with a dual acceleration strategy to improve the convergence speed of the algorithm. The dual acceleration strategy of the proposed optimization algorithm smooths and speeds up the originally steep, slow gradient descent in the parameter space. Compared with competing algorithms, this strategy improves the convergence speed of the algorithm by four times and the steady-state performance index by 96%. In addition, to verify the source separation performance of the algorithm more comprehensively, the simulation data with prior knowledge and the measured data without prior knowledge are used to verify the separation performance. Both simulation results and validation results based on measured data indicate that the new algorithm not only has better convergence and steady-state performance than conventional algorithms, but it is also more suitable for engineering applications.

## 1. Introduction

Electrical devices are widely applied and densely placed in marine integrated power systems. Owing to these characteristics, electromagnetic compatibility on site tests are susceptible to interference from the external environment, and these test results are often mixed with electromagnetic radiation from different devices. Regarding the testing of the electromagnetic radiation of devices in a system-level complex electromagnetic environment, one major problem to solve is obtaining the electromagnetic radiation signal of each device from the complex environment without a priori knowledge of such signals [1]. Blind source separation (BSS) is an important method for solving this problem. In general, existing algorithms can be classified into the following categories: information theoretic algorithms [2,3,4,5,6,7,8,9], algebraic algorithms [10,11], and canonical correlation analysis (CCA) methods [12,13]. Among them, blind separation algorithms based on information theory are usually adaptive online learning algorithms, which generally have better stability and convergence. Some blind separation algorithms based on second-order statistics, such as the algorithm for multiple unknown signals extraction (AMUSE) [14], and the second-order blind identification (SOBI) algorithm [15], use the non-zero time delay correlation function of the source signal to determine the separation matrix using matrix eigenvalue decomposition, which has the advantage of low computational effort and good stability for the source signal for any probability distribution. However, the decorrelation BSS algorithm requires the source signal to be completely uncorrelated for optimal performance; that is, the correlation matrix of the source signal should be a diagonal array. The SOBI algorithm can be regarded as an improvement of the AMUSE algorithm. The algorithm first requires pre-whitening of the observed signal, and this pre-whitening process is usually performed by the eigenvalue decomposition of the correlation matrix; however, this process reduces the separation performance of the algorithm. Such algorithms use optimization techniques, so that the algorithm performance will be better than that of the decorrelation algorithm. The BSS algorithm based on higher-order statistics [16] uses the higher-order statistical properties of the source signal to separate the signals. Such algorithms require, at most, one Gaussian signal in the source signal in addition to statistical independence of the source signal; this has the advantage of being resistant to Gaussian noise and allows the extraction of a single signal without involving other signals. This is another advantage over second-order statistics-based algorithms. The CCA method uses the autocorrelation function of the source signals to perform BSS. This has an advantage over the ICA-based separation method in that it not only considers the statistical distribution of the sample values but also makes full use of the correlation between the signals. In addition, CCA can eliminate the objective function constraint and the shortage of fixed-step gradient descent through the recursive framework. This ensures that the method can perform real-time data detection and source separation well even in highly damped systems.

The above-mentioned algorithms are divided into two categories in terms of implementation: batch and adaptive processing. Batch processing is based on a batch of previously acquired data, rather than the recursive processing of continuous input data. As this method is computationally tedious and ineffective, it is less frequently used in practical applications. Adaptive processing, such as the BSS algorithm based on the NPCA criterion and Kalman filtering [17,18], can gradually update the processor parameters with the acquisition of data one after another. The process uses real-time streams of data for monitoring and source separation, so that the application is broader than batch processing, and the computation is relatively simple. However, it experiences drawbacks of slow convergence and large steady-state errors. More promising neural network (NN)-based full optimization algorithms [19,20,21] do not require pre-whitening of observations, thus bypassing the problem of eigenvalue decomposition. This class of algorithms reduces the BSS problem completely to an optimization problem and, thus, has better separation performance than the first two classes of algorithms. From the perspective of NNs, the aforementioned BSS algorithm can be regarded as an optimization problem, i.e., the objective function is set to achieve the separation of mixed signals by finding the extreme value point. The objective function and optimization algorithm are crucial for NN solutions of the BSS problem. Different objective functions are constructed according to different criteria to form different BSS algorithms. For example, the SOBI algorithm [15] constructs the objective function using the joint diagonal method, and the NMF-based BSS [22] constructs the objective function by calculating the Kullback-Leibler (KL) divergence of *X* and AS (where *X* is the array of mixed signals, *A* is the mix matrix, and *S* is the array of sources). Any change in the objective function requires re-deriving the gradient descent formula, which is unfavorable for the application of algorithm improvement. In contrast, NNs provide a new idea: both the objective function and optimization algorithm are considered separately, allowing for the improvement of the objective function and optimization algorithm separately. The proposed BSS algorithm is based on the above-mentioned considerations, and its improvement methods for the objective function and optimization algorithm are, in general, equally applicable to the BSS that relies on other criteria. Thus, it improves the BSS algorithms from the perspective of NNs.

Additionally, the combination of the NN algorithms with the latest optimization algorithms can yield algorithms with a higher convergence rate and better steady-state performance facilitating the engineering application of BSS algorithms. For example, a mini-batch optimization algorithm [23] is used to improve the performance of BSS. However, the monotonous learning rate is usually too aggressive, and learning stops prematurely. Currently, the commonly used optimization algorithms for NNs are stochastic gradient descent and its variant the Nesterov accelerated gradient (NAG) [24], AdaGrad [25], Adadelta [26], RMSprop [27], and Adam [28]. The NAG (an improved momentum method) makes a correction during the gradient update, which accelerates the convergence and suppresses the oscillation. AdaGrad uses the cumulative sum of squared gradients as an accelerating variable; however, at the middle and late stages of training, the parameter update amount tends towards 0, resulting in a failure to continue learning. Adadelta adopts an exponential moving average of the sum of squared gradients, i.e., a closer current gradient corresponds to a larger gradient weight. In adadelta algorithm, the learning accelerates quickly at the early and middle stages but jitters around the local minimum at the late stage. RMSprop changes the cumulative sum of squared gradients of AdaGrad to an exponentially weighted moving average, thereby improving the aggressive and monotonously decreasing learning rate of AdaGrad. Adam incorporates the momentum directly into the estimation of the gradient first-order moment, which avoids the high bias of RMSProp at the beginning of training; consequently, the learning rate lies in a fixed range for each epoch, making the parameters comparatively smoother. There is a wide variety of optimization algorithms, but the dominant approach is to obtain the constraint term of the learning rate from the historical gradients to improve the convergence rate and steady-state performance. The foregoing optimization algorithms are not all applicable to BSS. In the case of BSS, there are numerous training samples. Gradient estimation through mini-batch gradient descent is even more inaccurate, which causes the training process to oscillate violently. Therefore, it is necessary to develop a more stable and reliable optimization algorithm. According to an analysis of the literature, we propose a new neural-network optimization algorithm that is suitable for BSS.

Many studies have proposed performance metrics, such as PI [29,30], $\zeta_{ij}$ [31], and signal-to-noise ratio (SNR) [32], for simulation experiments. These experiments and their evaluation metrics are useful for the study of algorithms because the simulated (or even experimental) signal carries information that accurately characterizes the system (i.e., the source signal and its statistical characteristics, and the system mixing matrix). This helps evaluate the model and improve the algorithm. For example, the source signal can be used to calculate the correlation of the separation result or estimate its probability density function. In contrast, the mixing matrix characterizes the coupling properties of the system and can be used to evaluate the estimated mixing matrix, or even evaluate the intermediate results of NN model training to obtain the dynamic properties of the algorithm. However, it is not suitable for engineering applications because, in practical applications, only the observed signal of the system can be obtained. Without a priori knowledge, such as the source signal and mixing matrix, the above performance metrics would be invalid. Studies on the evaluation indices of the separation performance of blind signals are lacking, making the engineering application of the algorithm difficult. Therefore, a performance index suitable for measuring the effect of BSS should be designed to guide the parameter adjustment of the algorithm and to evaluate the advantages and disadvantages of the algorithm.

In summary, the advantage of NNs for solving BSS problems is that the objective function and optimization algorithm are relatively independent. In engineering applications, the objective function and optimization algorithm can be combined with different criteria in any combination according to the actual demand, which is highly flexible. Second, the optimization algorithm of NN can overcome the problems of conventional BSS with a fixed step size, as well as batch processing, using a small batch variable step size algorithm. Based on the above-mentioned advantages of NNs and considering the slow convergence and low separation performance of conventional BSS methods in engineering applications [31], a BSS method that combines the maximum likelihood estimation (MLE) criterion and an NN with a bias term is proposed in this study. The method addresses problems in engineering applications in the following three ways:Improving the cost function for BSS: According to the MLE cost function, the training parameters of the feedforward NN (*W* and *b*) are added to the cost function as penalty terms through regularization to improve the learning performance.Improving the optimization algorithm of the NN: A dual-acceleration strategy is adopted. First, the cumulative historical gradient is obtained using the momentum term. Simultaneously, the learning rate is adaptively adjusted according to the modified exponentially weighted root mean square. The learning rate is increased at the early stage of training to ensure the convergence rate and reduced at the late stage of training to avoid excessive oscillation, so that the cost function in the parameter space descends rapidly along a flat path.A performance index suitable for engineering applications, i.e., ζ, is proposed. This index does not rely on source signals or the mixing matrix, making it useful for parameter adjustment in practical applications.

The remainder of this paper is organized as follows: In Section 2, the BSS method based on an NN with a bias term and the MLE criterion is briefly introduced according to three aspects: the NN structure, cost function, and optimization algorithm. In Section 3, the proposed algorithm is applied to the simulation data. The convergence speed and steady-state performance of the proposed algorithm are compared with those of conventional and real-time methods. In Section 4, the proposed algorithm is applied to actual test data to verify the separation performance of the algorithm in complex engineering applications. The conclusions are presented in Section 5.

## 2. Blind Source Separation Method Based on Neural Network (NN) with Bias Term and MLE Criterion

### 2.1. Neural Network (NN) Model with Bias Term

A neural-network model (as shown in Figure 1) based on the proposed algorithm was designed. The dense layer represents the fully connected layer, and the weight of this layer is the unmixing matrix of BSS to be solved. The lambda layer and loss layer jointly provide optimization objectives for the NN. The output of the loss layer is the objective function to be optimized by the NN. Similar to parts of the feedforward network, the lambda and loss layers do not involve training parameters, and their structures do not change with the optimization of the model. The aforementioned objective function contains a penalty term for the weight of the dense layer. In this manner, a regularization term for the weight is added to the loss function, which avoids model overfitting and allows for better BSS. The parameter settings of the NN are presented in Table 1.

The structure of the BSS method based on the MLE criterion and a self-organizing NN is shown in Figure 2. This algorithm uses the MLE criterion to achieve unsupervised learning in a feedforward NN with a bias term, and its learning criterion adopts an improved RMSprop optimization algorithm (see Section 2.3 for details), which can quickly and accurately converge to the global minimum by changing the MLE criterion to the negative log likelihood (NLL) loss (see Section 2.2), making it possible to estimate the mixing matrix *A* for BSS.

Compared with the conventional BSS methods, the self-organizing NN BSS method considers the BSS problem from the perspective of NNs, and can use the mature NN architecture Keras to implement the algorithm. Thus, it does not require the derivation of a parameter update formula by using the maximum likelihood function, Infomax (information maximization), minimum mutual information, or absolute value of the maximum kurtosis. Instead, these criteria must be converted into the regularization terms of the layer weights, layer biases, and layer outputs of the NN to obtain the loss function. Subsequently, the BP algorithm is used to learn *W* and *b* in the BSS algorithm. In addition, the learning criterion in the BSS+NN implementation is an improved RMSProp optimization algorithm, which does not require the participation of all samples for gradient computation. It is, therefore, less computationally intensive than conventional BSS algorithms. In summary, the proposed method has advantages over conventional BSS methods in terms of both implementation and computational complexity.

Additionally, the proposed algorithm is more scalable than the conventional methods. Because the computation of the loss function is relatively independent of the BP algorithm, various optimization algorithms can be applied to this structure to improve the algorithm performance.

### 2.2. Loss Function of Neural Network

In contrast to the common BSS algorithms based on the minimum mutual information criterion, the proposed algorithm introduces a bias term in the feedforward NN, subsequently regularizes the bias term, and adds it to the loss function. The bias term then serves as a training parameter in model optimization. This prevents overfitting in training and solves the problem of the loss not decreasing in the neural-network optimization process.

The following log-likelihood function (loss function) for neural-network optimization is based on the MLE criterion:(1)$$\begin{array}{cc} \ell (W,b) & =\log|\det\left(W\right)|+\sum_{i=1}^{N}E\left\{\log g_{i}^{\prime }\left(Y_{i}\right)\right\}\\ \; & -E\left\{\log p_{X}\left(X\right)\right\} \end{array}.$$

The constant $-E\left\{\log p_{X}\left(X\right)\right\}$ can be discarded because it does not contribute to the training of the NN. Considering that the NN training takes the minimum of the objective function as the optimization direction, the maximum likelihood objective function takes a negative value. In practical computation, the sample mean is the expected unbiased estimate; thus, the objective function ℓ(W,b) is given as follows:(2)$$\begin{array}{cc} \ell (W,b) & =-\log|\det\left(W\right)|-\sum_{i=1}^{N}E\left\{\log g_{i}^{\prime }\left(Y_{i}\right)\right\}\\ \; & ≃-\log|\det\left(W\right)|-\frac{1}{m}\sum_{i=1}^{N}\sum_{j=1}^{m}\log g_{i}^{\prime }\left(Y_{ij}\right), \end{array}$$
where *m* and *N* represent the sample points and dimensions of the observed signal $Y_{i}$, respectively; $g_{i}\left(•\right)$ and ${g^{\prime }}_{i}\left(•\right)$ represent the probability distribution function and the probability density function (PDF) of the source signal *i*, respectively; *W* represents the to-be-optimized weight of the NN, i.e., the unmixing matrix of the BSS problem to be solved; and the bias term *b* is implicitly included in $Y_{i}$. The PDF can be selected based on the distribution type of the source signals, as displayed in Table 2. Figure 3 shows the negative logarithmic probability density estimation functions for different distributions.

The L2 regularization term is added to the loss function given by Equation (Equation 2) as a penalty term to avoid overfitting in the optimization. Therefore, the proposed neural-network loss function ℓ(W,b) comprises two parts: the NLL cost and the L2 regularization cost. The impact of the latter part on the gradient manifests in the weight decay of the gradient:(3)$$\begin{array}{cc} \ell (W,b) & =\underset{NLL\;\mathrm{cost}}{\underset{︸}{-\log|\det\left(W\right)|-\frac{1}{m}\sum_{i=1}^{N}\sum_{j=1}^{m}\log g_{i}^{\prime }\left(Y_{ij}\right)}}\\ \; & +\underset{L2\;\mathrm{regularization}\;\mathrm{cost}\;}{\underset{︸}{\frac{1}{m}\frac{\lambda }{2}\left({∥W∥}_{2}^{2}+{∥b∥}_{2}^{2}\right)}} \end{array},$$
where *m* represents the sample points, and λ is the weight factor of the regular term, which essentially controls the weight decay of *W* and *b*. Namely, the weight is reduced by a quantity proportional to *W* by λ. When λ=0, ℓ(W,b) degenerates to the NLL cost. In addition, if λ is too large, the negative logarithmic likelihood loss insignificant. To obtain the appropriate hyperparameter λ, this study uses the GridSearchCV API in Scikit-learn (Machine Learning in Python) [33,34] to debug the parameters of the NN.

### 2.3. Improved Optimization Algorithm

After the loss function of the NN is obtained, the model parameters *W* and *b* can be learned using the back propagation (BP) algorithm. The proposed neural-network optimization algorithm is an improved version of the RMSprop optimization algorithm [25], as indicated by Equation (Equation 4).
(4)$$\begin{array}{cc} \; & r\left(w,t\right)=\rho r\left(w,t-1\right)+\left(1-\rho \right){\left(\nabla Q_{i}\left(w\right)\right)}^{2}\\ \; & w=w-\frac{\eta }{\sqrt{r(w,t)+\epsilon }}\nabla Q_{i}\left(w\right) \end{array},$$
(5)$$\begin{array}{cc} \; & r\left(w,t\right)=\rho r\left(w,t-1\right)+\left(1-\rho \right){\left(\nabla Q_{i}\left(w\right)\right)}^{2}\\ \; & g\left(w,t\right)=\rho g\left(w,t-1\right)+\left(1-\rho \right)\nabla Q_{i}\left(w\right)\\ \; & v\left(w,t\right)=\beta v\left(w,t-1\right)+\frac{\eta }{\sqrt{r\left(w,t\right)-{\left(g(w,t)\right)}^{2}+\epsilon }}\nabla Q_{i}\left(w\right)\\ \; & w=w-v(w,t) \end{array}.$$

Here, ρ represents the decay rate of the exponential moving average; β is the momentum term; ϵ is a minimal constant, which avoids divide-by-zero errors in the update process; η represents the global learning rate; $\nabla Q_{i}\left(w\right)$ represents the gradient at the current time *t*; g(w,t) represents the estimate of the (exponentially weighted) first-order moment of the gradient; and r(w,t) is the gradient accelerating variable. RMSprop uses the decay average of the previous squared gradient (second-order moment) instead of the value.

As indicated by Equation (Equation 5), in this study, the standard RMSprop optimization algorithm is improved with regard to two aspects:The momentum term is introduced to accumulate the previous gradients for accelerating the current gradient.g(w,t), i.e., the estimate of the first-order moment of the gradient, is introduced. The original r(w,t) in RMSprop is modified to the central second-order moment through the operation $r\left(w,t\right)-{\left(g(w,t)\right)}^{2}$. In order to stabilize the exponentially weighted root mean square, this operation flattens the steep gradient in the parameter space. In practice, the algorithm finds a smoother descent direction in the parameter space, increasing the training speed.

To qualitatively analyze the optimization process of the improved algorithm, we plotted the direction, step size, and descent process of the gradient at each step of iteration for the common optimization algorithms and the improved algorithm, as shown in Figure 4. The direction of stochastic gradient descent (SGD) roughly represents the direction of the gradient at the current position. Although the algorithms can finally reach the target point through stepwise descent along the gradient direction, the entire descent process oscillates violently, and the descent is slow. The update direction at each step is not optimal in the proposed algorithm. However, the introduction of the momentum term ensures gradient inertia and overcomes the unevenness of changes among different training parameters, making the whole descent process smoother and reducing the oscillation of the descent process. The descent speed of Adagrad is slow, although its optimization direction is similar to that of the improved algorithm. The improved algorithm ensures rapid descent at the early stage and slower descent at the late stage of training through gradient accumulation, which allows the optimization process to converge quickly and smoothly in the target direction.

To quantitatively verify the effectiveness of the proposed optimization algorithm, we compare the improved optimization algorithm presented in this paper with the commonly used optimization algorithms reported in the literature [35]. The parameter settings of each optimization algorithm are presented in Table 3.

Figure 5 shows the final performance of each algorithm after their parameters are repeatedly debugged. We conclude that compared to commonly used optimization algorithms, the proposed optimization algorithm can ensure comparatively faster convergence at the early stage of the training process and can span multiple local minima of the loss function; it has a higher descent rate at both the early and mid-late stages of the training process. It is more suitable for dealing with non-smooth targets. The proposed optimization algorithm has a higher descent rate at the early stage because of its dual-acceleration strategy. The previous gradients are accumulated using the momentum term. At the early stage, the centralized exponentially weighted root mean square $\sqrt{r\left(w,t\right)-{\left(g(w,t)\right)}^{2}}$ is small; thus, the learning rate of the global η increases significantly. With increasing training epochs, $\sqrt{r\left(w,t\right)-{\left(g(w,t)\right)}^{2}}$ increases gradually to ensure a decline in the global learning rate, preventing oscillation at the global optimum.

As shown in Figure 6, the figure shows the weight update rate of each gradient of each algorithm. As can be seen from the graph, the optimization algorithm designed in this paper can advance to the goal with a large range, and the amplitude of the whole learning process is kept in a high range and decreases slowly. The NAG algorithm in the initial training update range is too large, it is too radical, and the medium update range is too small, easy to fall into local optimum, and not easy to find the global optimum. The other algorithms in the initial training update range is relatively small, indicating that the algorithm update process is conservative and convergence speed will be relatively a small, comprehensive comparison, and the proposed algorithm update range of the whole training process is moderate, both to ensure the convergence speed and stability.

In summary, the proposed algorithm combines an NN with a bias term, a loss function with a weight and bias penalty, and an improved optimization algorithm (Figure 2). The detailed flow of the algorithm is shown in Algorithm 1. It is worth noting that the loss function is highly nonconvex, and, if the initial value is determined randomly according to the conventional algorithm, then each algorithm is unstable and may obtain a different local minimum each time. This is very unfavorable for algorithm tuning and practical applications. The above-mentioned analysis of the algorithm clearly shows that the parameter optimization process of the algorithm can be divided into two parts:the parameters of Equation (Equation 3), that is, λ, $W_{0}$, $b_{0}$, which determine the starting point of the cost function optimization; andthe optimization algorithm and its parameters, which determine the descent path of the cost function ℓ(W,b).
**Algorithm 1:** BSS based on an NN with a bias term and the MLE criterion **Input**: Dataset comprising observation channels: data_tf; Training epochs: epochs=5000; Sample batch size: batch_size=200; Regularization function for *W* and *b*: tf.keras.regularizers.l2(); Constructed neural-network model: creat_model(); Performance index: PI as shown in Equation (Equation 9); Performance index: ξ, as shown in Equation (Equation 10); **Output**: Estimated source signal dataset: estimated_data**1** Define the NN inputs InputLayer and DenseLayer based on the data samples;**2** Randomly initialize the weight matrix *W* and bias term *b*;**3** Design the neural-network loss function my_loss according to Equation (Equation 3);**4** Construct a neural-network model based on the proposed algorithm (Figure 2)  create_model(w_reg=tf.keras.regularizers.l2(),b_reg=tf.keras.regularizers.l2());**5** Import data into the training network, and record the weight *W* and bias term *b* during training;**6** Calculate the performance index PI=my_metrics(),ξ=episilion() for each step using *W* and *b* in the process;**7** Return the estimated signal source dataset estimated_data;

Therefore, to ensure that each operation of the algorithm can obtain stable and reliable separation results, the following steps were considered according to the optimization process of the algorithm. First, initial values $W_{0}$, $b_{0}$ were fixed using random seeds or debugging experience to ensure that the algorithm has the same starting point for each run of the loss function. Next, all hyperparameters of the whole optimization process were searched using GridSearchCV with PI as the evaluation index grid. Finally, the optimal hyperparameters with PI<0.02 are extracted. According to the above-mentioned strategy, this not only avoids the instability of the algorithm caused by the uncertainty of the initial values but also improves the algorithm’s optimization search path to obtain suitable separation results through the hyperparameter search.

## 3. Simulation Analysis of Algorithm Performance

This section validates the proposed algorithm using simulation data with a priori knowledge (i.e., the source signal waveform and mixing matrix are known). The advantage of simulation data validation is that a priori knowledge is available, and the separation results of the algorithm can be evaluated by calculating the performance metrics associated with a priori knowledge. The simulation validation in this study is divided into six parts:First, the real-time algorithm in the recursive framework is used as a competing algorithm to verify the computational performance of the proposed algorithm.The second part focuses on the adjustment of the algorithm hyperparameters and the adaptation capability of the algorithm under different numbers of samples and sources.The third part verifies the influence of the distribution type of the source signal on the performance of the algorithm.The fourth part verifies the impact of the added regularization term on the performance of the algorithm.The fifth part comprehensively compares the performance of the proposed algorithm with that of the conventional algorithm through several performance metrics.The sixth part verifies the separation performance of the proposed algorithm for sparse data.

We set the source signal as shown in Equation (Equation 6).
(6)$$S=\left\{\begin{array}{c} s_{1}=\sin\left(2t\right)\\ s_{2}=sign\left(\sin\left(3t\right)\right)\\ s_{3}=sawtooth\left(2\pi t\right) \end{array}.\right.$$

The source signal parameters are set as follows: *t* represents the time series that is determined when the number of sampling points is N=2000 and the sampling frequency is $f_{s}=125$ Hz. Here, sign(•) can be expressed by Equation (Equation 7). The sawtooth wave $s_{3}$ has a time period of 1. Its value increases from −1 to 1 at the time interval from 0 to 1 and then decreases from 1 to −1 at the time point 1.
(7)$$sign\left(x\right)=\left\{\begin{array}{c} -1,x<0\\ 0,x=0\\ 1,x>0 \end{array}.\right.$$

The source signal mixing matrix *A* is randomly generated by the standard normal distribution, as given by Equation (Equation 8). The Gaussian white noise is added to the source signal at a certain SNR to verify the noise immunity of the proposed algorithm.
(8)$$A=\left[\begin{array}{ccc} 1.3316 & 0.7153 & -1.5454\\ -0.0084 & 0.6213 & -0.7201\\ 0.2655 & 0.1085 & 0.0043 \end{array}\right].$$

For the application scenario of this study, the performance metric (Equation (Equation 9) is customized) to measure the similarity between the global matrix and the identity matrix. The metric is based on the literature [30] for improving the generic performance metric [29], which is essentially a normal form of the literature [30]. A smaller PI corresponds to a better separation effect. When PI=0, the waveform of the separated signal perfectly matches that of the source signal.
(9)$$\begin{array}{cc} PI(G) & =PI(BA)\\ \; & =\frac{1}{n}\sum_{i=1}^{n}\left(\sum_{j=1}^{m}\frac{{\left|g_{ij}\right|}^{2}}{\max_{k}{\left|g_{ik}\right|}^{2}}-1\right)\\ \; & +\frac{1}{m}\sum_{j=1}^{m}\left(\sum_{i=1}^{n}\frac{{\left|g_{ij}\right|}^{2}}{\max_{k}{\left|g_{ki}\right|}^{2}}-1\right) \end{array}$$

Here, $g_{ij}$ is an element in the global matrix G=BA, where *B* represents the unmixing matrix, and *A* represents the mixing matrix. $\max_{k}{\left|g_{ik}\right|}^{2}$ denotes the maximum value of the square of the element in the *j*th column of *G*, and $\max_{k}{\left|g_{ki}\right|}^{2}$ denotes the maximum value of the square of the element in the *i*th row of *G*. *k* represents the index of the row/column to which the maximum element corresponds. However, in most cases, PI [36] is only applicable to the measurement of the separation effect of the known simulated signals in the mixing matrix *A*. Therefore, we introduce the correlation coefficient as a performance index [37] to evaluate the advantages and disadvantages of the separation algorithm.
(10)$$\xi_{ij}=\xi \left(y_{i},s_{j}\right)=\left|\sum_{t=1}^{k}y_{i}\left(t\right)s_{j}\left(t\right)\right|/\sqrt{\sum_{t=1}^{k}y_{i}^{2}\left(t\right)\sum_{t=1}^{k}s_{j}^{2}\left(t\right)}.$$

Here, $\xi_{ij}$ represents the similarity coefficient between the *i*th column of the separated signal *y* and the *j*th column of the source signal *s*; and *k* represents the number of sampling points of the data. Here, $0\leq \xi_{ij}\leq 1$; when $y_{i}=cs_{j}$ ( *c* is a constant), $\xi_{ij}=1$. When $y_{i}$ and $s_{j}$ are independent of each other, $\xi_{ij}=0$. A larger $\xi_{ij}$ corresponds to a larger similarity coefficient between $y_{i}$ and $s_{j}$ and a better separation effect. The similarity coefficients of the proposed algorithm and the conventional MLE-based ICA algorithm are presented in Table 4. As shown, the similarity coefficients of the components in the proposed algorithm are >0.9975, indicating that the proposed algorithm exhibits a better separation effect than the conventional MLE-based ICA algorithm.

### 3.1. Computational Performance Verification of the Proposed Algorithm

To verify the performance of the proposed algorithm in terms of computation, AMUSE, SOBI, FOBI, and RCCA implemented in the recursive framework were used as competing algorithms in this study; this avoids matrix inversion operations and is simpler to compute compared with batch computation. In addition, the online real-time recursive implementation of the competitive algorithms helps view the optimization process of the unmixing matrix and facilitates the comparison of the proposed algorithms. The convergence rate is also used as an indicator to compare the performance of the above algorithms with the proposed algorithm. Figure 7a shows the change in performance metrics during the iterations of the proposed and competing algorithms, which reflects the convergence speed and steady-state performance of the different algorithms. As shown in Figure 7a, although the competing algorithms implemented in the recursive framework are computationally simpler, their overall convergence speed is inferior to that of the RMLE-ICA algorithm with a dual acceleration strategy, and the proposed algorithm RMLE-ICA has a smaller PI in terms of steady-state performance; this proves that the proposed algorithm is more thorough in separating the mixed signals. While SOBI, an improved AMUSE algorithm, has improved steady-state performance, it has reduced convergence speed. Figure 7b shows the convergence speed of different algorithms when they reach the steady state, characterized by the number of iterations of the algorithm at convergence. As shown in the figure, the convergence speed of the proposed algorithm in this study is significantly better than that of the competing algorithms, which is mainly owing to the fact that the proposed algorithm uses a dual acceleration strategy in the optimization part to better balance the convergence speed and steady-state performance. In summary, the proposed algorithm has better convergence speed and steady-state performance than the competing algorithms in the recursive framework.

### 3.2. Algorithm Hyperparameter Tuning and Performance Verification

In this section we focus on the impact of hyperparameters batch_size, and λ on the performance of the algorithm and how they can be adjusted. In addition, we verify the adaptability of the proposed algorithm to the number of samples *m* and the number of source signals *N*, as well as the effect of increasing the number of samples *N* on the running time of the algorithm.

The essence of the optimization algorithm of the NN is gradient descent. At present, gradient descent adopts the mini batch gradient algorithm. However, the gradient is easy to run off, which affects the performance of the algorithm. Therefore, in order to select the appropriate method, this study compares the different effects on the convergence and steady-state performance of the algorithm. As shown in Figure 8, when the epoch is fixed, the smaller batch_size converges faster, while in the steady state, PI has little difference with different value of batch_size and is essentially below 0.01. This shows that a smaller batch_size not only has a fast convergence speed but also has good steady-state performance.

Considering that parameter λ in Equation (Equation 3) has a great impact on the performance of the algorithm, this study adopts GridSearchCV in Scikit-learn to adjust λ. GridSearchCV is essentially an exhaustive method, that is, the selection of all candidate parameters is traversed by a loop, trying every possibility, and the parameter with the best evaluation index is the final result. The evaluation index used in this study is PI. The PI for different λ values is shown in Figure 9. Intuitively, when λ=0.01, the entire training process of the algorithm not only converges quickly but also separates most thoroughly. Of course, the selection of λ is not the smaller the better. When λ=0, the regular term in the loss function ℓ(W,b) fails, and, although the algorithm can converge, the performance of the algorithm is not the best. In summary, to obtain the best algorithm performance, in this study, λ was set as 0.01.

Next, we compare the steady-state separation performance of the proposed algorithm with those of the MLE-based ICA, L1 regularization algorithm for different sampling points *m*. The experimental results are shown in Figure 10a, which shows the steady-state separation performance of all three algorithms is improved to different degrees with an increase in the number of sample points. This could mainly be because when the number of sample points increases, the number of sample points used for gradient calculation at the current position of the loss function increases, thus yielding more accurate gradient estimates and causing the algorithm to converge to the optimal position along a more suitable path. In summary, the steady-state performance of both the L1 regularization and proposed algorithms is significantly better than that of MLE-ICA. The small difference between the steady-state performances of the L1 regularization and proposed algorithms indicates that the regularization term mainly improves the convergence speed of the algorithm in the pre-training period and has little effect on the steady-state performance of the algorithm. In addition, the increase in sample points does not significantly improve the steady-state performance of the algorithm; this indicates that the proposed algorithm can achieve better performance even with smaller samples.

In additon, we also investigated the effect of *m* on the running time of the algorithm. The running times of different algorithms with different values of *m* are shown in Figure 10b. As shown in the figure, the training time of the three algorithms gradually increases as the number of sample points increases; this is because the increase in samples leads to an increase in the number of samples involved in the calculation of the gradient at the current position, and this in turn increases the running time of the algorithms. Thus, we can conclude that the improvement in the separation effect is at the cost of the training time.

Finally, we explored the impact of increasing the number of sources *N* on the performance of the algorithm. The number of source signals is increased by changing the frequency of $s_{1}$ while keeping $s_{2}$ and $s_{3}$ of Equation (Equation 6) constant. The test results are shown in Figure 11. As shown, among the three algorithms, only the algorithm proposed in this paper can guarantee that the performance index PI is always less than 0.2 as the number of source signals increases, and the algorithm can be considered to be successful in engineering applications. The separation results of the other two algorithms are not stable, and the L1 algorithm can only separate successfully at the number of individual source signals. Therefore, the algorithm proposed in this paper can adapt to the change in the number of source signals and can obtain satisfactory separation performance for both large and small numbers of sources.

### 3.3. Influence of Distribution Type of Source Signal on Algorithm Performance

In the derivation of the loss function of the neural algorithm, the greatest influence on the whole optimization process is the negative logarithmic probability density estimation function $-\log g_{i}^{\prime }\left(Y_{ij}\right)$ in Equation (Equation 3). Most application scenarios assume that $g_{i}\left(•\right)$ is the sigmoid function in $-\log g_{i}^{\prime }\left(Y_{ij}\right)$, where the default source signal distribution is a Gaussian distribution. However, the probability density distribution of the actual signal does not necessarily satisfy this assumption. Only when the probability density of the signal is close to the estimated probability density can the algorithm achieve better separation performance [38]. Considering that the signal characteristics of the source signal in the simulated signal are known, the proposed algorithm adopts the following flow to process the simulated signal.
Calculate the statistical indicators (skewness, kurtosis) of the simulation signal.Determine the general distribution type to which the source signal belongs by means of statistical indicators.Select the corresponding $g^{\prime }\left(s\right)$ from Table 2 according to the distribution type.

Next, we process the simulated signal as described above. The distribution indicators (skewness, kurtosis) are shown in Table 5, which shows that the kurtosis of the simulation signal is negative (the kurtosis of the normal distribution is 0), indicating that the source signal is a sub-Gaussian distribution. Accordingly, the probability density function of the source signal is obtained by kernel density estimation method (kernel = ‘Gaussian’). Figure 12 also proves that the source signal is more consistent with the characteristics of the sub-Gaussian distribution.

Next, we discuss the influence of the probability distribution type of the estimated source signal on the performance of the algorithm from two aspects:The algorithm employs different PDFs corresponding to those in Table 2 for the simulated signals of the sub-Gaussian distribution, aiming to verify the performance of the proposed algorithm under the model mismatch condition.The algorithm for selecting the sub-Gaussian distribution is compared with ICA-EBM [9] and ICA-EMK [38] to verify the effectiveness of the proposed algorithm.

The performance indices of the three $g^{\prime }\left(s\right)$ models for the separation results of the simulated signal where the source signal is sub-Gaussian are shown in Figure 13. By comparison, it is found that the super-Gaussian distribution model cannot successfully separate the simulated signal and oscillates severely. In contrast, both the Gaussian distribution model and the sub-Gaussian distribution model can achieve accurate separation of the source signal (steady-state PI close to 0). The difference is that the sub-Gaussian distribution model has a faster convergence speed. In other words, the Gaussian distribution model mismatch of the algorithm proposed in this paper only affects the convergence speed of the algorithm without affecting the separation results, proving that the Gaussian distribution model has certain generalization ability, which is the reason why the algorithm takes it as the first choice.

From the above analysis, it is clear that for the simulated signal, the sub-Gaussian distribution model is the best choice. To further verify the performance of the proposed algorithm, ICA-EBM and ICA-EMK are used as competing algorithms, and the average correlation coefficient and PI are used as performance indicators, in this study. Figure 14 presents the separation performance of the proposed algorithm with those of ICA-EBM and ICA-EMK for the simulated signals. As far as the average correlation coefficient is concerned, the performances of the three algorithms do not differ significantly. However, from the perspective of PI, the proposed algorithm separates more thoroughly than the competing algorithms. Therefore, it can be concluded that the proposed algorithm is better than the competing algorithms ICA-EMK, and ICA-EBM.

Both ICA-EMK and ICA-EBM algorithms are based on the maximum entropy principle, which requires the mean, variance, and higher order statistics of the source signal to be known in order to estimate the probability density function of the source signal more accurately, which is often difficult to satisfy in practice. The algorithm proposed in this paper, however, only requires a general understanding of the distribution type of the source signal to obtain a more desirable separation result, which proves that the proposed algorithm is more practical.

### 3.4. Impact of the Regularization Term on Algorithm Performance

To verify the effect of the regularization term introduced in the NN loss function on the BSS, we used PI as the performance index to apply the algorithm without a regularization term, that is, the L1 regularization algorithm and the L2 regularization algorithm to the simulated signals. The PI changes that occurred during the training process are plotted in Figure 15. The parameters of the three aforementioned algorithms were fixed to avoid the effects of other factors on the training results, as shown in Table 6.

As shown in Figure 15, all three algorithms experienced fluctuations in the early stage. With increasing epochs, the performance index (PI) decreases, and the separation effect of the BSS algorithm improves, finally stabilizing at a certain value. The algorithms with a regularization term performed better than the algorithm without a regularization term in terms of both the convergence rate and steady-state performance. The proposed algorithm converged to <0.01 after 414 epochs and finally converged to approximately 0.0073. The L1 regularization algorithm converged to <0.01 after 1664 epochs and finally converged to approximately 0.0035. The algorithm without a regularization term failed to converge to <0.01 and finally stabilized near 0.1946. In summary, the algorithm without a regularization term failed to achieve the actual separation performance and fell into the local optimum. The L1 regularization algorithm had a smaller steady-state performance index, but its convergence rate was nearly four times higher than that of the proposed algorithm. The proposed algorithm yielded good results with regard to both the convergence and steady-state performance; compared with the MLE-ICA algorithm, the convergence speed is increased four-fold, and the steady-state performance index is improved by 96%. Thus, it is more suitable for situations requiring a short training time. This proves that the addition of a regularization term can significantly improve the separation effect of the BSS algorithm.

### 3.5. Comparison of Multi-Index with Traditional Algorithm

Considering that the prior knowledge (i.e., the mixing matrix *A*, the source signal *s*) of the simulation signals is known, we further verify the separation performance of the algorithm by applying Corr (Equation (Equation 11)) between the source signal *s* and separation signal $\hat{s}$.
(11)$$Corr\left(s_{j},{\hat{s}}_{j}\right)=\frac{1}{m}\sum_{j=0}^{m-1}\frac{\sum \left(s_{j}-m_{s_{j}}\right)\left({\hat{s}}_{j}-m_{{\hat{s}}_{j}}\right)}{\sqrt{\sum {\left(s_{j}-m_{s_{j}}\right)}^{2}\sum {\left({\hat{s}}_{j}-m_{{\hat{s}}_{j}}\right)}^{2}}},$$
where *s* and $\hat{s}$ are both n×m dimensional signal matrices, *n* is the sample points, *m* is the signal dimension, $s_{ij}$ is the *j*th column of the *i*th row of the matrix, and $s_{j}$ is the *j*th column of the matrix.

The proposed algorithm is compared with four competitive ICA algorithms: FastICA, JADE density function (PDF) [39,40], CuBICA [41], and TDSEP [42]. JADE is a cumulant-based batch algorithm for source separation, and we use the $N^{2}$ version in the comparisons. FastICA is based on entropy approximation, and we use the symmetric decorrelation approach. CuBICA is an improved cumulant-based batch-algorithm that is able to handle linear mixtures of symmetrically and skew-symmetrically distributed source signal components. TDSEP is an online algorithm based on only time lagged second order correlations, i.e., it is suited to be trained on huge data sets, provided that the training is done sending small chunks of data each time. The key parameters of the above algorithm are configured as shown in Table 7.

The results are shown in Figure 16. From the perspective of PI, the performance of the algorithms is similar except for TDSEP, where CuBICA and the proposed algorithm perform the best (PI < 0.02). However, when Corr is considered, the average correlation coefficient between the separation results of the proposed algorithm and the source signal is closer to 1 than that of CubICA, which indicates that the separation signal of the proposed algorithm is closer to the source signal. In summary, both performance metrics of the proposed algorithm are the best (PI = 0.007, Corr = 0.999), which fully demonstrates the better steady-state separation performance of the proposed algorithm compared with the competing algorithms.

### 3.6. Separation Performance of the Proposed Algorithm for Sparse Data

The ability of the proposed algorithm to separate common source signals has been verified above, and the performance of the proposed algorithm to separate sparse source signals is further verified in this study. In this paper, three sparse sources, $s_{1}$, $s_{2}$, and $s_{3}$, composed of smooth bell-shaped signals (Equation (Equation 12)), are selected, and the sources are approximately independent. The composition of each sparse source signal is shown in Table 8. The mixing matrix *A* of the source signals is consistent with the previous *A* (Equation (Equation 8)).
(12)$$s\left(t\right)=A_{0}e^{a(t-\tau )},a>0,$$
where *t* is the time series, $A_{0}$ controls the signal amplitude, a=10 controls the width of the bell signal, and τ is the signal offset. The source signal obtained from the above-mentioned parameters is shown in Figure 17. Next, the source signal is superimposed with Gaussian noise (μ=0,σ=0.01), and the observed signal matrix *X* is obtained by mixing the matrix *A*.

For the sparse simulation data, three methods (MLE-based ICA, L1 regularization algorithm, and the proposed algorithm) are used to process the sparse simulation data. The optimal optimization parameters of the proposed algorithm are listed in Table 9. Finally, the learning curve of the performance index PI with the training process epoch is shown in Figure 18, where both the MLE-based ICA and L1 regularization algorithm move towards the decreasing direction of the cost function; however, they fluctuate significantly, and are finally limited to the local minimum value. On the contrary, because of the existence of a penalty term, the proposed algorithm can select a more suitable gradient at the beginning of training, and with the appropriate optimization algorithm, it can converge to the ideal position (PI < 0.1) quickly and accurately, which can meet the needs of engineering applications. From the above-mentioned analysis, we conclude that the proposed algorithm is more suitable for sparse data processing than the conventional algorithms.

## 4. Verification of Algorithm Performance for Actual Data

To verify the separation effect of the proposed algorithm in practical engineering applications, we applied the proposed algorithm to the source signal separation task in the field environment of the generator set in a real ship’s engine room. The test equipment and the field test environment are shown in Figure 19. The log-periodic antenna is distributed along the generator set. The mixed signal spectrum collected by the sensor is shown in Figure 20a. The frequency information of the source signal is difficult to distinguish. A correlation analysis was first conducted to evaluate the correlations among all the channels, as shown in Figure 20b. The three component frequencies with the highest correlations were *f* = [91.8 MHz, 435 MHz, 623 MHz], with their corresponding correlations being cxy=[0.625,0.709,0.394]. Then, the measured signal was processed by applying the MLE algorithm [4] and the proposed algorithm.

The separation performance indices for the existing algorithms [32], such as the PI and $\xi_{ij}$, are only suitable for evaluating the BSS performance of the systems with known source signals or mixing matrices. However, the above conditions are unknown in actual tests. To evaluate the separation performance, we designed a performance index ζ for actual tests.
(13)$$\zeta ={∥\xi \left(Y,X\right)-\hat{I}∥}_{2}.$$

Here, $\xi \left(Y,X\right)$ represents the matrix of similarity coefficients between the separated signal *Y* and the observed signal *X*, $\hat{I}$ represents the identity matrix of $\xi \left(Y,X\right)$, and ${∥•∥}_{2}$ represents the second norm of the matrix. Because the separated signal is a component of the observed signal, there is a correlation between them. Larger separation corresponds to greater similarity between the signals, i.e., a smaller ζ corresponds to a better separation effect. The value of ζ cannot be 0; ζ=0 indicates that the separated signal is identical to the observed signal.

Similarly, taking ζ as the performance index, we employed the MLE-based ICA algorithm, L1 regularization algorithm, and proposed algorithm for the BSS of real signals. The changes in the performance index ζ during the training for all the algorithms are presented in Figure 21. As shown, the performance index of the standard MLE-based ICA algorithm remained relatively unchanged, indicating that this algorithm was poor for separating real signals. The L1 regularization algorithm and the proposed algorithm both exhibited a decline in ζ, indicating that the algorithm played a role in improving the separation effect. Overall, the proposed algorithm had the best separation effect but also converged rapidly. The final values of ζ are presented in Table 10. According to the results, we conclude that the proposed algorithm is more suitable than the other algorithms for engineering applications.

Figure 22 shows the correlation coefficients between the separation results and the observed signals for each algorithm. IC2 exhibited a larger correlation coefficient with OB0 and a smaller correlation coefficient with OB1 for the proposed algorithm than for the other algorithms, indicating that the proposed algorithm contains fewer components related to OB2. This also suggests that the proposed algorithm had a more thorough separation result. In summary, the proposed algorithm has a more thorough separation result and better BSS performance than the other algorithms in practical engineering applications.

The separation result of the proposed algorithm is shown in Figure 23. Comparing Figure 23a,b reveals that the separation of the MLE algorithm is not thorough and that the components of the source signal are difficult to distinguish, whereas the proposed algorithm can clearly find the components of the observed signal *f* = [91.8 MHz, 435 MHz, 623 MHz]. This observation is consistent with the results of the previous correlation analysis, which proves the effectiveness of the proposed algorithm in practical engineering applications.

## 5. Conclusions

A BSS method combining an NN with a bias term and the MLE criterion is proposed in this paper. The introduction of the L2 regularization of the bias term in the NN avoids overfitting during the training of the NN and improves the convergence rate and steady-state performance of the algorithm. Additionally, an optimization algorithm with a dual-acceleration strategy for neural-network learning is presented. This strategy employs the momentum term and the modified exponentially weighted root mean square to accelerate the gradient. It has stronger adjustment inertia compared with conventional optimization algorithms and can make the originally steep gradient direction in the parameter space smooth and fast. A simulation analysis of the proposed algorithm confirmed its anti-interference ability and its advantages over the MLE-based ICA with regard to the convergence rate and steady-state performance. Furthermore, the proposed algorithm has a simple structure, without the whitening (pre-processing) of conventional BSS algorithms. The algorithm is highly scalable and, thus, can be continuously improved to enhance its performance. However, there are limitations related to the implementation of the algorithm. Because the learning rate in the optimization of the NN is still relatively fixed, a learning-rate regulator should be introduced in the future to dynamically adjust the learning rate of the optimization algorithm for improving its performance. Additionally, in future research, we expect that NNs will be used to estimate the number of signal sources.

## Figures and Tables

**Figure 1 sensors-21-00973-f001:**
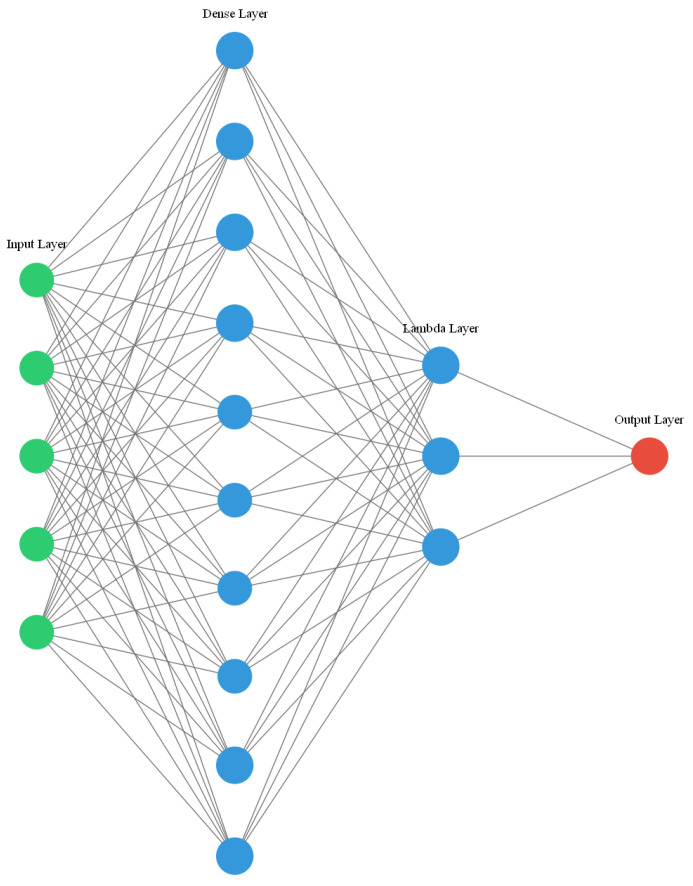
Structure diagram of a neural network (NN).

**Figure 2 sensors-21-00973-f002:**
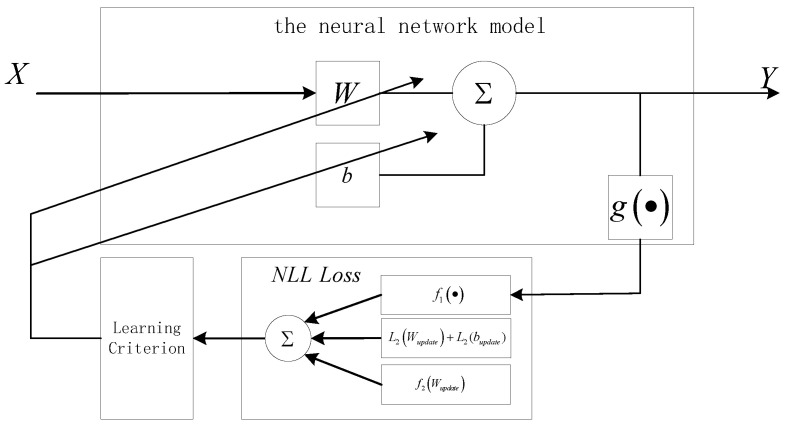
Structure diagram of the blind source separation (BSS) method based on the maximum likelihood estimation (MLE) criterion and a self-organizing NN.

**Figure 3 sensors-21-00973-f003:**
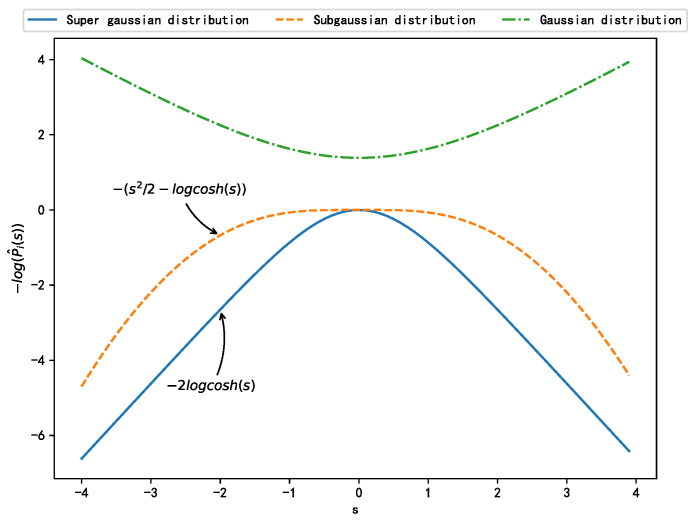
Estimation of negative logarithmic probability density corresponding to different distributions.

**Figure 4 sensors-21-00973-f004:**
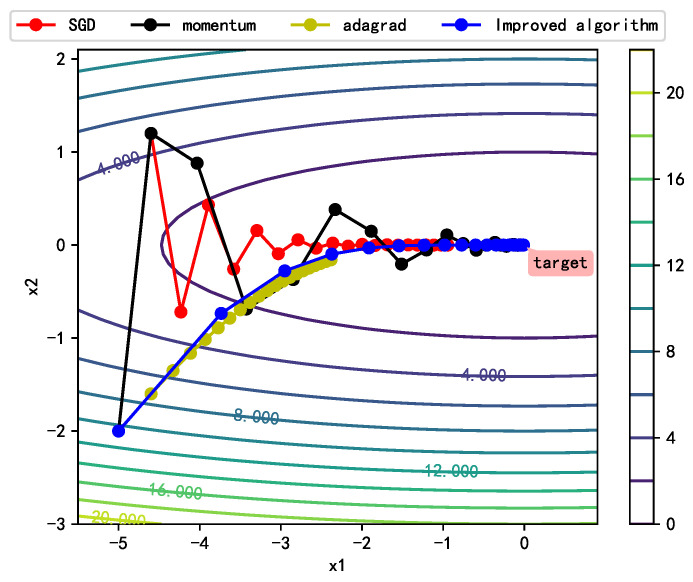
Changes of the gradient vector in the contour curve.

**Figure 5 sensors-21-00973-f005:**
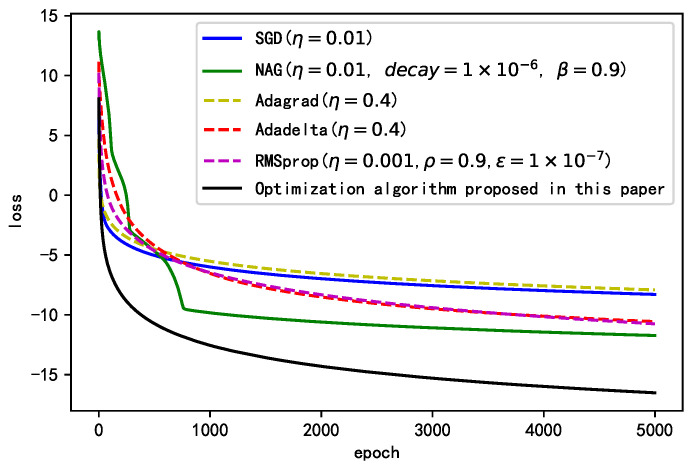
Learning curves of different optimization algorithms.

**Figure 6 sensors-21-00973-f006:**
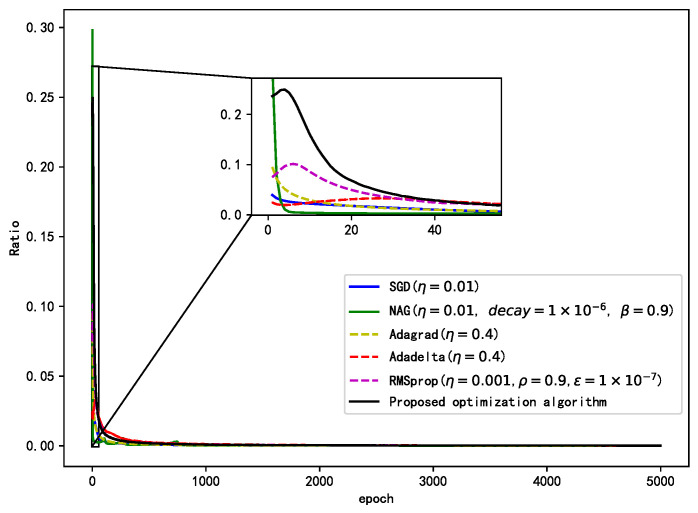
Weight update rate of different optimization algorithms.

**Figure 7 sensors-21-00973-f007:**
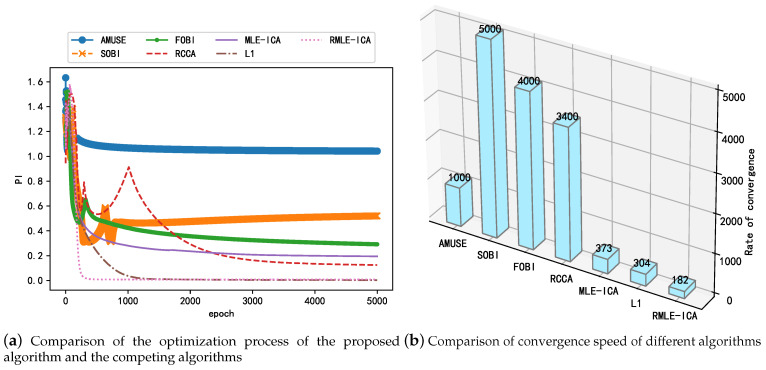
Comparison of computational performance between the proposed algorithm and competing algorithms.

**Figure 8 sensors-21-00973-f008:**
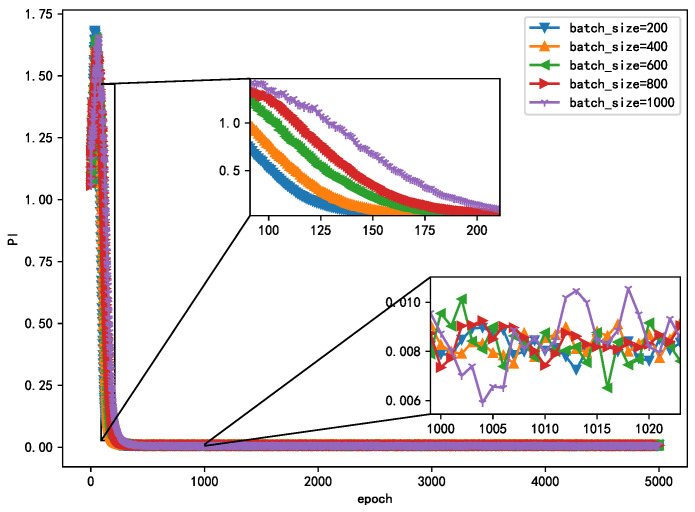
Learning curves of different batch_size algorithms.

**Figure 9 sensors-21-00973-f009:**
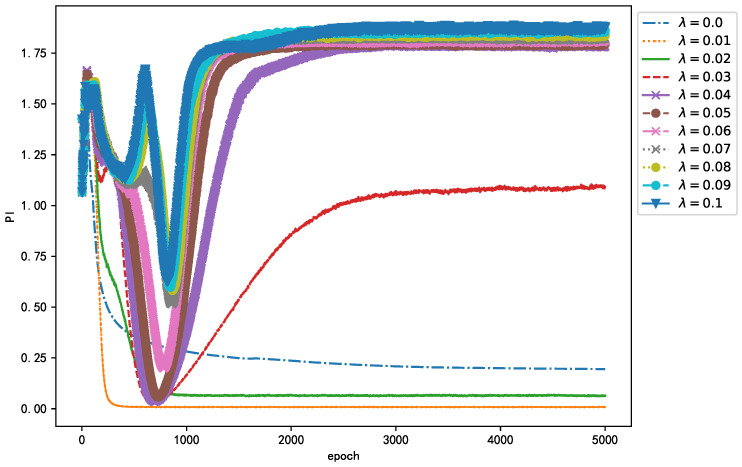
Learning curves for different values of λ.

**Figure 10 sensors-21-00973-f010:**
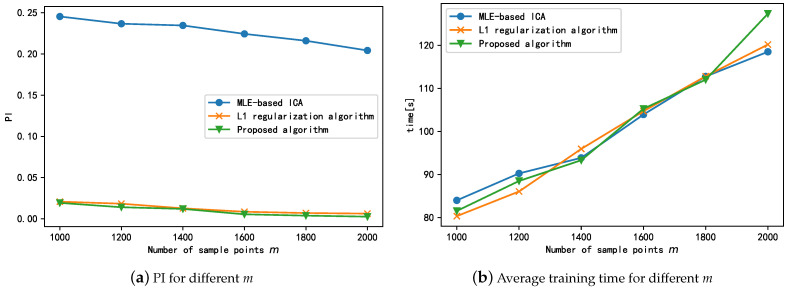
Effect of sampling points *m* on algorithm performance and training time.

**Figure 11 sensors-21-00973-f011:**
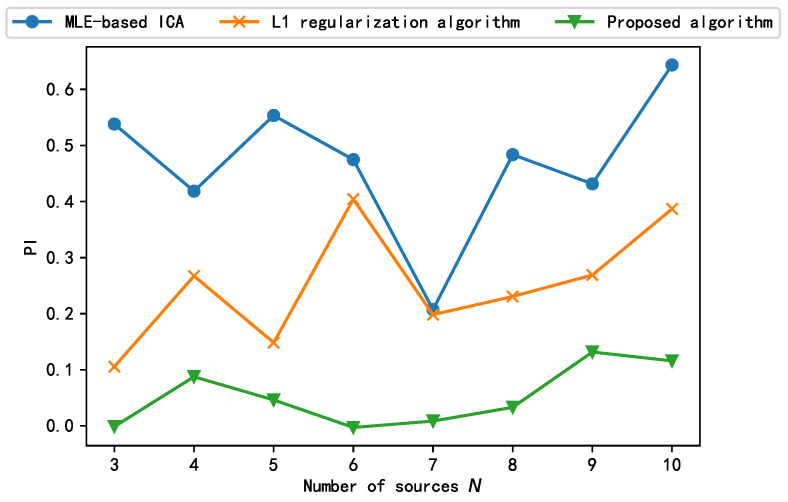
Separation performance of proposed algorithm and comparison algorithm under different number of sources.

**Figure 12 sensors-21-00973-f012:**
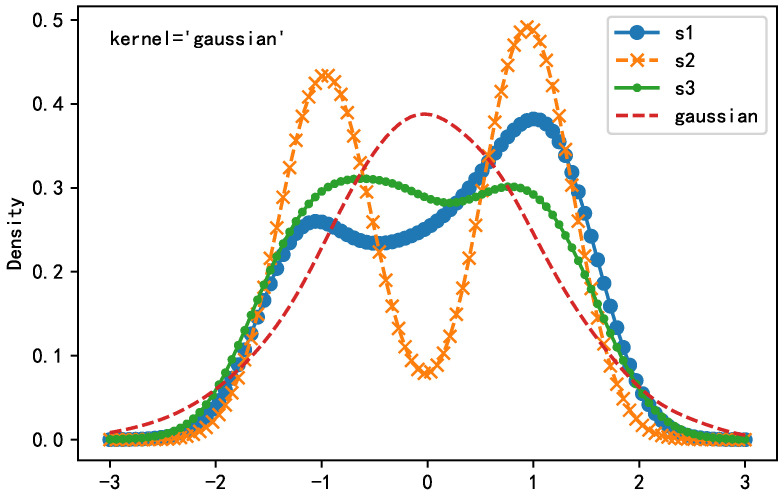
Kernel density estimation of source signals.

**Figure 13 sensors-21-00973-f013:**
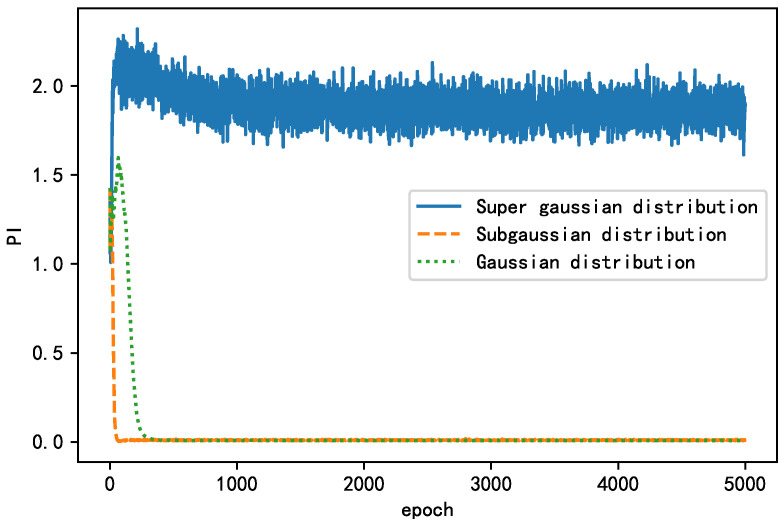
Performance index of corresponding algorithm with different PDFs.

**Figure 14 sensors-21-00973-f014:**
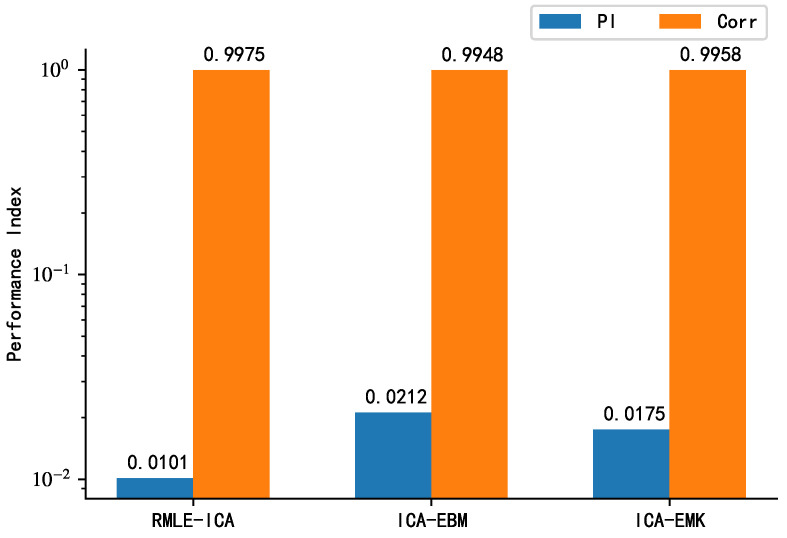
Performance comparison of proposed algorithm with ICA-EBM density function (PDF) and ICA-EMK density function (PDF) for sub-Gaussian distribution of source signals.

**Figure 15 sensors-21-00973-f015:**
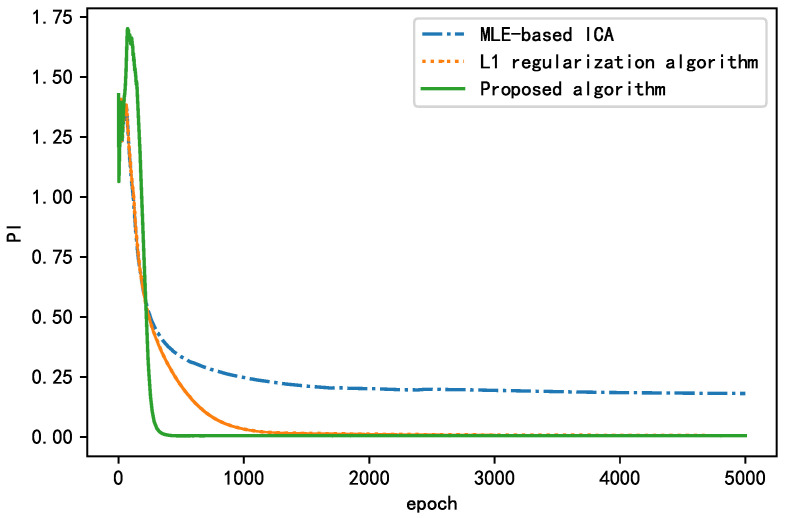
Comparison of learning curves between the proposed algorithm and MLE-ICA.

**Figure 16 sensors-21-00973-f016:**
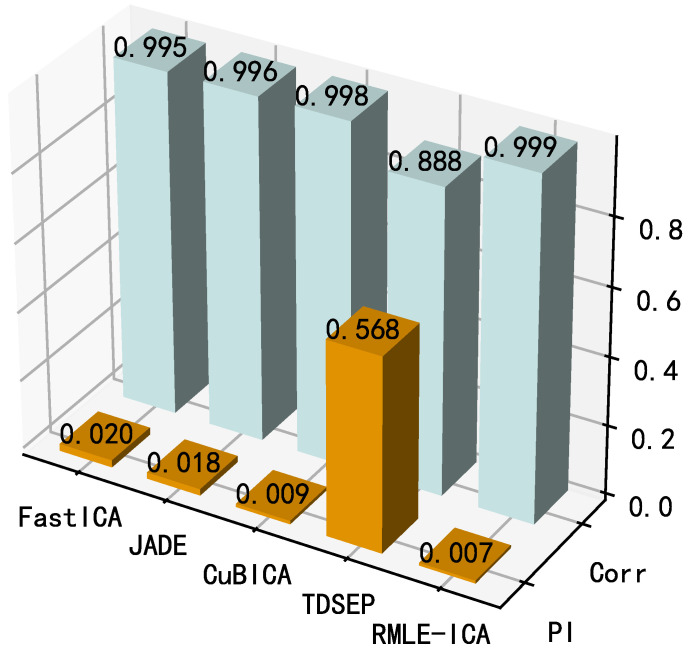
Comparison of steady-state performance between proposed algorithm and conventional algorithms.

**Figure 17 sensors-21-00973-f017:**
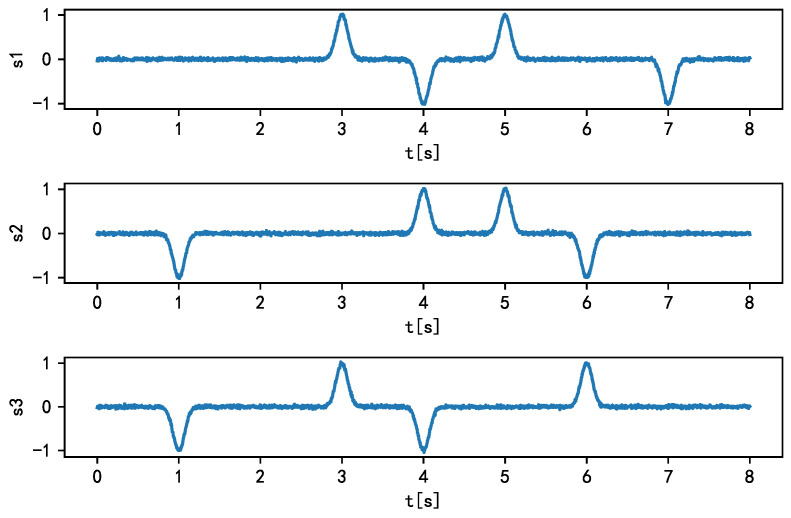
Time-domain waveform of source signal.

**Figure 18 sensors-21-00973-f018:**
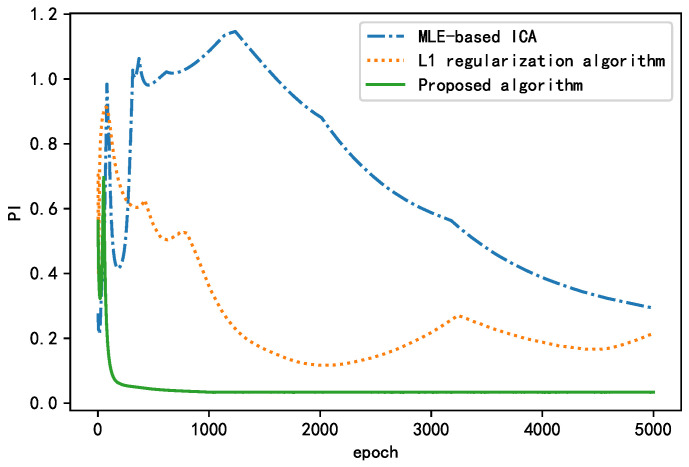
Performance comparison of different algorithms for sparse data separation.

**Figure 19 sensors-21-00973-f019:**
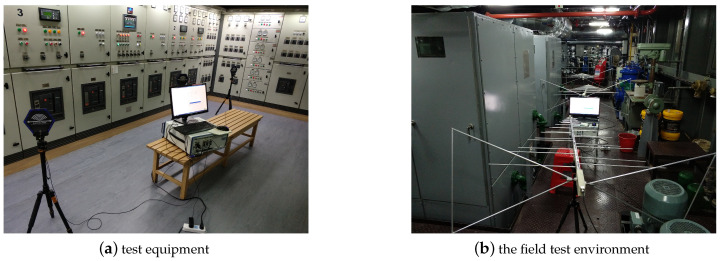
Field environmental test of generator set in ship engine room.

**Figure 20 sensors-21-00973-f020:**
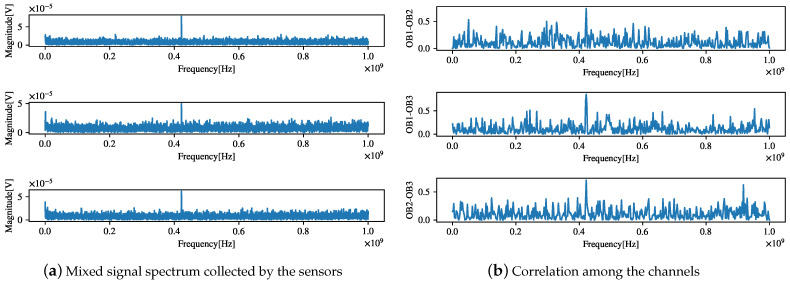
Mixed signal spectrum collected by the sensor and correlation analysis.

**Figure 21 sensors-21-00973-f021:**
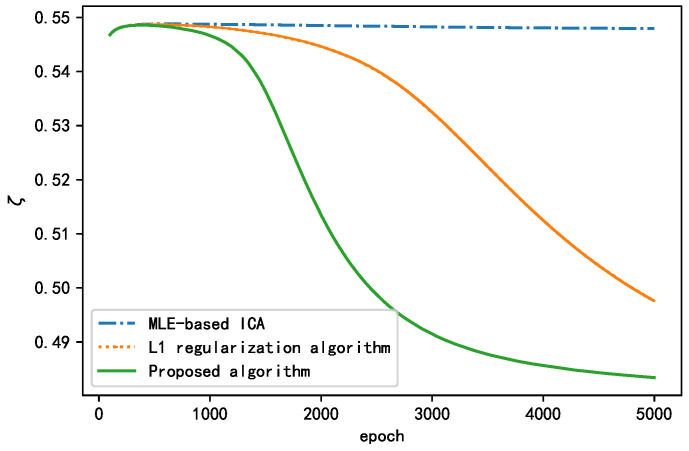
Changes in during the training process for each algorithm.

**Figure 22 sensors-21-00973-f022:**
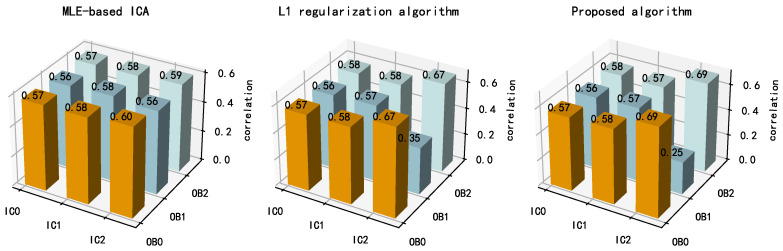
Correlation coefficients between the separation results and the observed signals for each algorithm.

**Figure 23 sensors-21-00973-f023:**
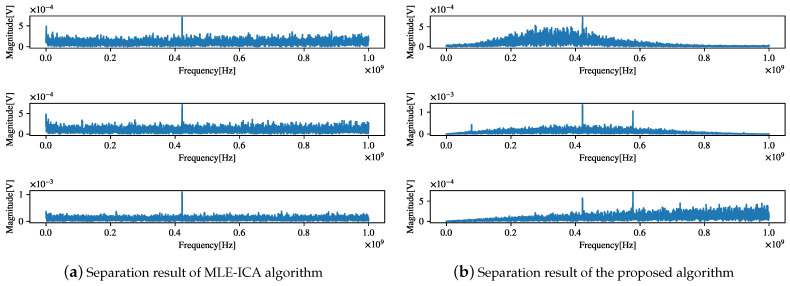
Separation results of real-ship tests.

**Table 1 sensors-21-00973-t001:** Parameter settings of the neural-network model.

Neural-Network Layers	Parameters
Input layer	Input dimensions: *n*
Dense layer	Output dimensions: *m* Activation function: sigmoid Trainable parameters: *W* and *b* *W* regularization: Loss1 *b* regularization: Loss2
Lambda layer	Fuzzy factor: ε=1 × 10${}^{-8}$ Output: Loss3
Loss layer	Output: Loss1+Loss2+Loss3

**Table 2 sensors-21-00973-t002:** Probability density function (PDF) corresponding to different distributions.

Distribution Type	Function Expression
Super Gaussian distribution	$g^{\prime }\left(s\right)=-2logcosh\left(s\right)$
Sub-Gaussian distribution	$g^{\prime }\left(s\right)=-\left(s^{2}/2-logcosh\left(s\right)\right)$
Gaussian distribution	g′(s)=gs1−gs,g(s)=11(1+exp(−s)(1+exp(−s))

**Table 3 sensors-21-00973-t003:** Different optimization algorithms and their parameter settings.

Optimization Algorithm	Parameter Settings
SGD	η=0.01
NAG	η=0.01,β=0.9,decay=1 × 10${}^{-6}$
Adagrad	η=0.4
Adadelta	η=0.4
RMSprop	η=0.001,ρ=0.9,ϵ=1 × 10${}^{-7}$
Proposed optimization algorithm	η=0.01,ρ=0.9,β=0.9,ϵ=1 × 10${}^{-7}$

**Table 4 sensors-21-00973-t004:** Similarity coefficients of the proposed algorithm and the conventional MLE-based ICA algorithm.

	Proposed Algorithm	MLE-Based ICA
ξ	$\left[\begin{array}{ccc} 0.0259 & 0.9978 & 0.0083\\ 0.9980 & 0.0230 & 0.0117\\ 0.0733 & 0.0593 & 0.9975 \end{array}\right]$	$\left[\begin{array}{ccc} 0.9823 & 0.0275 & 0.0767\\ 0.0535 & 0.9989 & 0.0185\\ 0.0739 & 0.0164 & 0.9968 \end{array}\right]$

**Table 5 sensors-21-00973-t005:** Distribution index of simulation signals.

	Source Signal *S*			Observation Signal *X*		
	$s_{1}$	$s_{2}$	$s_{3}$	$x_{1}$	$x_{2}$	$x_{3}$
Skewness	−0.252	−0.094	0.026	−0.15	−0.02	−0.16
Kurtosis	−1.153	−1.640	−1.027	−0.506	−0.701	−0.927

**Table 6 sensors-21-00973-t006:** Model and training parameters.

Model and Training Parameters	Description
batch_size=200	Number of samples involved in gradient calculation in the optimization algorithm
epochs=5000	Number of times the full sample is processed during training
step=N/batch_size=10	Number of training steps per epoch
η=0.01,ρ=0.9,β=0.9,ϵ=1 × 10${}^{-7}$	Hyperparameters of the proposed optimization algorithm

**Table 7 sensors-21-00973-t007:** Parameter setting of contrast algorithm.

	Algorithm Parameter
FastICA	approach = ‘defl’, g = ‘pow3’, mu = 1, max_it = 5000, limit = 0.001
JADE	limit = 0.001, max_it = 1000
CuBICA	limit = 0.001
TDSEP	lags = 1, limit = 1 × 10${}^{-5}$, max_iter = 10,000

**Table 8 sensors-21-00973-t008:** Parameter configuration of sparse source signals.

Source Signal	Parameter Combination
$s_{1}$	$A_{0}=\left[1,-1,1,-1\right],\alpha =\left[3,4,5,7\right]$
$s_{2}$	$A_{0}=\left[-1,1,1,-1\right],\alpha =\left[1,4,5,6\right]$
$s_{3}$	$A_{0}=\left[-1,1,-1,1\right],\alpha =\left[1,3,4,6\right]$

**Table 9 sensors-21-00973-t009:** Optimal parameters for algorithm debugging determined by *GridSearchCV*.

Adjustable Term	Optimal Parameters
the cost function ℓ(W,b)	$W_{0}=tf.keras.initializers.RandomNormal\left(seed=1\right)$ $b_{0}=tf.keras.initializers.RandomNormal\left(seed=1\right)$ λ=0.01
Optimization algorithm	epoch=5000 batch_size=150 η=0.01,ρ=0.9,β=0.9,ϵ=1 × 10${}^{-7}$

**Table 10 sensors-21-00973-t010:** Performance indices for each algorithm at the end of the training.

	MLE-Based ICA	L1 Regularization	Proposed Algorithm
ζ	0.548	0.497	0.483

## Data Availability

Data can be available on request to authors.

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
