# Peer review of "Blind Source Separation Method Based on Neural Network with Bias Term and Maximum Likelihood Estimation Criterion"

_sensors, 2021, doi:10.3390/s21030973_

Round 1

Reviewer 1 Report

Authors propose a novel technique for solving the blind source separation problem that combines a maximum likelihood cost and a neural network with a bias term. 

This is an interesting piece of work however, there are some important points that authors need to address before this article can be considered for publication. 

Comments: 

  1. On equation (3) authors consider the sigmoid function as the probability density for the estimated sources Y_i. It is not clear why this is a reasonable assumption to make and what is its impact on the separation performance. It has been shown that separation performance of ICA highly depends on the estimated probability density of the latent sources [1]. Based on this how do authors justify the selection of (3)? Is this selection ideal for a wide range of applications? Is there a physical meaning?
  2. Based on the previous comment what happens to the separation performance of the algorithm if there is a model mismatch? More experiments need to be considered and authors need to compare with a wide range of ICA algorithms similarly to [1],[2].
  3. How sensitive is algorithm on the selection of \lambda in (4)? How do authors perform hyperparameter tuning? Please describe through extra experiments/discussion.
  4. What is the impact on the average CPU time of the proposed algorithm if you have a large number of sources N to estimate? How does algorithm scale for large N and m? Also is there a significant affect on the separation performance if you have small sample size?
  5. Does the proposed algorithm work for sparse data? It would be of high interest to consider a small study regarding this point. Similar study can be found here [3].
  6. Please use the correct references for equation (10). References 23,24 are not the correct ones. 
  7. It looks that equation (12) is not the correct equation to consider when comparing ICA algorithms. It is known that ICA can estimate W up to a scaling ambiguity. Therefore, MSE is not a fair metric to consider. 
  8. How do authors consider the most stable or most consistent run for each algorithm in their experiments? It is known that cost functions are highly non-convex so different local minima can be obtained every time the algorithm runs. Please discuss.
  9. Please proof-read article. There are several places that need minor corrections.

[1] Boukouvalas, Zois, et al. "Independent component analysis using semi-parametric density estimation via entropy maximization." 2018 IEEE Statistical Signal Processing Workshop (SSP). IEEE, 2018.

[2] Li, Xi-Lin, and Tülay Adali. "Independent component analysis by entropy bound minimization." IEEE Transactions on Signal Processing 58.10 (2010): 5151-5164.

Author Response

Dear Reviewer of the Sensors.

Subject: Response to reviewer.

I am writing to you regarding the reply of the submitted paper titled “Blind Source Separation Method Based on Neural Network with Bias Term and Maximum Likelihood Estimation Criterion”. In the attached file you can find a specific reply for each comment of the reviewers. Furthermore, the additional clarifications in the new document submitted are marked in yellow color.

Thank you for taking time for reading our article and for your detailed comments. On behalf of all co-authors, we sincerely hope the revised version of the paper has been much improved to reviewer’s satisfaction.
On behalf of all the authors,
Bangmin Wang, corresponding author

Reviewer 2 Report

Please find my comments in the attached document. 

Author Response

(The authors gave the same response as above.)

Round 2

Reviewer 1 Report

The results of the updated paper look very reasonable and meaningful. Authors have incorporated all of my feedback and have done a very good job with new experiments and discussions given the limited time-frame. 

I do not have additional comments. 

Author Response

Dear Reviewer of the Sensors.

Subject: Response to reviewer.

I am writing to you regarding the reply of the submitted paper titled “Blind Source Separation Method Based on Neural Network with Bias Term and Maximum Likelihood Estimation Criterion”. Thank you very much for your approval of our modification.

Thank you for taking time for reading our article and for your detailed comments.

On behalf of all the authors,
Bangmin Wang, corresponding author

Reviewer 2 Report

Although the authors have addressed most of my queries, there are some aspects that have been ignored and misrepresented. These issues are enumerated with respect to the queries sent out in the previous query sheet: 

  1. Query 3: The authors have chosen the methods accurately to further the cause in this context. However, lines 33-36 are completely misinterpreted in the sense that BSS based recursive canonical correlation analysis, SOBI, FOBI, AMUSE, Kalman filtering, and diagonalization can all be carried out in an online, real-time, recursive mode. The authors should justify this with a proper narrative. 

2. Query 4: The use of CCA in a recursive framework alleviates the use of: 

a) objective function constraints

b) fixed steps gradient descent. 

In light of the performance of recursive canonical correlation analysis - using data that streams in real-time for monitoring and source separation - highly damped systems can be put under scrutiny to exhibit proper convergence results. This should be reflected in the paraphrase. 

3. Query 11: The authors should focus on the computational advantage of the proposed method in terms of either of the following: 

a) Floating-point operations per second (flops) 

b) Number of operations per iteration 

c) Rate of convergence. 

Also, in the context of this draft, it is desired that the method is compared against both traditional and real-time algorithms. 

4. Query 14: The 'noise immunity' context of the algorithm should be derived in terms of mathematical fundamentals. For instance, an algorithm like recursive singular spectrum analysis can filter out noise by reconstructing the time series only from the signal subspace and leaving the noise component behind. With applications in source separation - especially in structural health monitoring - the algorithm performs using the output from a single sensor and inherently embeds online filtering, without the use of a traditional filter. The performance of the proposed method could be compared to such benchmarks. 

Author Response

(The authors gave the same response as above.)
